# Vertical Motion of Air over the Indian Ocean and the Climate in East Asia

**Rongxiang Tian** [1,2,*], **Yaoming Ma** [3,4,5] **and Weiqiang Ma** [3,4,5]

1   School of Earth Sciences, Zhejiang University, Hangzhou 310027, China
2   Key Laboratory of Geoscience Big Data and Deep Resource of Zhejiang, Hangzhou 310027, China
3   Institute of Tibetan Plateau Research, Chinese Academy of Sciences, Beijing 100101, China; ymma@itpcas.ac.cn (Y.M.); wqma@itpcas.ac.cn (W.M.)
4   CAS Center for Excellence in Tibetan Plateau Earth Sciences, Chinese Academy of Sciences, Beijing 100101, China
5   College of Earth and Planetary Sciences, University of Chinese Academy of Sciences, Beijing 100049, China
\*   Correspondence: trx@zju.edu.cn

**Abstract:** The Indian Ocean and East Asia are the most famous monsoonal regions, and the climate of East Asia is affected by the change in wind direction due to monsoons. The vertical motion of the atmosphere is closely related to the amount of precipitation in whichever particular region. Climate diagnosis and statistical analysis were used to study the vertical motion of air over the Indian Ocean and its relationship with the climate in East Asia. The vertical motion of air over the Indian Ocean had a significant correlation with the climate in China—especially with precipitation in the Tibetan Plateau and the Yangtze River Basin—as a result of the interaction of the vertical motion of air from the Indian Ocean, the Tibetan Plateau and the subpolar region in the Northern Hemisphere. The vertical motion over the Indian Ocean was weakened from 1981 to 2010, except at a height of 500 hPa in winter. The vertical motion of air over the Indian Ocean had a period of 7–9 years in summer and 9–12 years in winter. An ascending motion was dominant over most of the Indian Ocean throughout the year and the central axis of the ascending motion changed from a clockwise rotation from winter to summer to a counterclockwise rotation from summer to winter as a result of the monsoonal circulation over the Indian Ocean. These results will provide a theoretical reference for a comprehensive understanding of the climate in Asia and contribute to work on climate prediction in these regions.

**Keywords:** Indian Ocean; East Asia climate; vertical motion of air; Tibetan Plateau

## 1. Introduction

The Indian Ocean and East Asia have a monsoonal climate [1]. The Indian Ocean is surrounded by continental regions. There are different radiation and energy balances between the land and ocean as a result of the different natures of the underlying land and ocean surfaces (e.g., the heat capacity and albedo, etc.), and these balances change with the seasons. Different pressure fields are produced in different seasons, which generate a seasonal change in the wind field, namely, monsoons [1,2]. The summer monsoon brings abundant precipitation, while the winter monsoon brings drought and little rain. In terms of the influence of the change in wind direction due to the monsoon on the climate, there has been much research concerning the influence of change in the horizontal wind field on climate, while the relationship between climate and change in the vertical wind field has not received enough attention. The vertical motion of air is the result of both thermal and dynamic action [3], and can be directly linked to precipitation. Heating of the underlying surface can be expressed directly as the vertical motion of air [4]. The characteristics and intensity of vertical motion in the atmosphere are closely related to precipitation. Precipitation is associated with updrafts and drought is associated with

downdrafts [5]. For example, on a temporal scale, short-term vertical motion is associated with small-scale precipitation—such as typhoons (hurricanes) and rainstorms—whereas long-term vertical motion is associated with large-scale rainy weather (mostly low-pressure areas) and droughts (mostly high-pressure areas) [6]. On a spatial scale, areas of ascent in the Hadley circulation correspond to low pressure and a rainy zone in the meridional direction, such as the region of ascending air near the equator, whereas areas of descending air correspond to high pressure and arid zones (e.g., the subtropical arid region) [7,8]. The area of ascent related to the Walker and anti-Walker circulations [9] corresponds to the rainy zone around the equator, whereas the area of descending air corresponds to regions with only small amounts of rain in the zonal direction [10,11].

The upward motion of air over the Indian Ocean and the downward motion of air over the surrounding continents form many vertical circulations [12] which affect the amount of precipitation. Bjerknes [13] studied the relationship between the vertical motion of the atmosphere and precipitation over India as early as 1910 and showed that the vertical motion of air over India is closely related to the amount of precipitation in the northern side of the Indian Ocean. On the western side of the Indian Ocean, the vertical motion of air over Africa is also closely related to the local precipitation, and has led to droughts in Africa during the last century [14,15]. Since China is on the path of the Indian monsoon, there has been much research on the influence of change in the horizontal wind field on climate in China [16,17], while the research on the relationship between the vertical movement of air from the Indian Ocean and the climate of faraway China is rare.

We investigated the effects of the vertical motion of air over the Indian Ocean on the climate in East Asia. We aimed to find out whether the vertical motion of air associated with the Indian Ocean monsoonal circulation plays a role in the formation of and change in the climate of East Asia.

The research results are of significance to understand comprehensively the climate of East Asia and make more accurate climate forecasts. The paper is organized as follows: Section 2 describes the data and methods. The results and discussion are presented in Sections 3–5. The study concludes with a brief summary in Section 6.

## 2. Data and Methods

### 2.1. Data

Data for the monthly mean vertical wind speed were obtained from the National Centers for Environmental Prediction/National Center for Atmospheric Research (NCEP/NCAR) reanalysis dataset [18] with a resolution of $2.5° \times 2.5°$. Twelve pressure levels were used for the Indian Ocean (1000, 925, 850, 700, 600, 500, 400, 300, 250, 200, 150 and 100 hPa). Precipitation data from 839 meteorological stations were provided by the China Meteorological Administration. The surface air temperature and atmospheric pressure were extracted from the Scientific Data Center for the Cold and Arid Regions of China surface meteorological datasets with a temporal and spatial resolution of $0.1° \times 0.1°$. The monthly mean data which were calculated by using these datasets form the basis of the analytical approach.

Taking into consideration the remote connection between the equatorial Indian Ocean's sea surface temperature and the East Asian climate [19], the location of the Indian Ocean was taken as (25° S–30° N, 20° E–125° E) and the location of China as (15° N–55° N, 70° E–145° E). The time period measured was 1981–2010. Due to the horizontal wind at 850 hPa, the vertical motion of air (omega) at 500 hPa and the sea surface temperature play an important role in some regions in the Indian Ocean, while outgoing longwave radiation and the vertical motion of air (omega) at 500 hPa dominate for other regions in the occurrence of extreme rainfall [20]. The sea surface temperature anomaly (dipole event) over the Indian Ocean also has a good correlation with the geopotential height of 500 hPa, and is closely related to the precipitation anomaly in China during summer [21,22]. Therefore, the vertical motions of 500 hPa and 850 hPa over the Indian Ocean are selected. We studied the correlation between vertical movement over the Indian Ocean in January as

well as June and the climate in East Asia, since the Indian Ocean summer monsoon erupts in June while January is the beginning of winter [23].

To determine the reliability of the data, we compared the vertical motion of air in the NCEP, ERA-Interim (produced by the European Center for Medium-Range Weather Forecasts) and JRA-55 (from the Japan Meteorological Agency) datasets over the Indian Ocean and the Tibetan Plateau and found that they had almost identical systems and centers [24]. It is therefore reasonable to analyze the vertical motion of air using the NCEP data.

### 2.2. Methodology

To diagnose and analyze the vertical motion of air over the Indian Ocean, we used empirical orthogonal function (EOF) analysis [25,26] to decompose the vertical velocity in winter (December–February), summer (June–August), January and June. The vertical velocity fields were decomposed into products of space function and time function by EOF decomposition. EOF analysis can be used to decompose the original data field, anomaly field and standardization field of vertical velocity. The results of decomposing different data fields are different in climatic significance. Because we performed orthogonal function decomposition on the original vertical velocity field, the first eigenvector (the spatial distribution) represents the average state (main pattern) of the vertical velocity field in the study area (explanation variance is large), and the corresponding time coefficient represents the time variation characteristics of the main pattern. The calculation of the EOF is as follows:

$$X_{M \times N} = V_{M \times P} \times T_{P \times N} \tag{1}$$

where $X_{M \times N}$ is a data matrix of the original vertical velocity composed of $N$ observations of $M$ spatial points. $V$ are eigenvectors and T are eigenvalues. We used the first eigenvector in the analysis. To determine whether the first eigenvector has a physical meaning, we used the rule suggested by [26] to test the results:

$$e_j = \lambda_j \left(\frac{2}{N}\right)^{\frac{1}{2}} \tag{2}$$

where $e_j$ is the error range of the eigenvalue $\lambda_j$ and $N = 30$ is the sample size. When the adjacent eigenvalues satisfied $\lambda_j - \lambda_j + 1 \geq e_j$, we considered that the EOFs corresponding to these two eigenvalues were significant.

The time series of the corresponding main pattern in winter and summer was used to extract the periodic variation signals of the spatial distribution pattern using Morlet wavelet analysis [27–29].

The continuous wavelet transform $W_n^X(s)$ on a scale s of a discrete time series $x_n$ ($n = 1, \ldots, N$) with uniform time steps $\delta t$ was defined as the convolution of $x_n$ with the scaled and translated version of the wavelet function $\psi_0$:

$$W_n^X(s) = \sqrt{\frac{\partial t}{s}} \sum_{n'=0}^{N-1} x_{n'} \psi_0^* \left[\frac{(n' - n)\partial t}{s}\right] \tag{3}$$

where * indicates the complex conjugate, $N$ is the total number of data points in the time series and $(\partial t/s)^{1/2}$ is the factor used to normalize the wavelet function, such that every wavelet function has a unit energy at each wavelet scale s.

By transforming the wavelet scale $s$ and localizing along the time index $n$, we obtained a diagram showing the fluctuation characteristics of the time series at a certain scale and its variation with time—that is, the wavelet power spectrum [27,28,30]. The Morlet wavelet is not only non-orthogonal, but is an exponential complex-valued wavelet regulated by a Gaussian distribution defined as:

$$\psi_0(t) = \pi^{-1/4} e^{i\omega_0 t} e^{-t^2/2} \tag{4}$$

where $t$ is the dimensionless time and $\omega_0$ is the dimensionless frequency. When $\omega_0 = 6$, the wavelet scale s is basically equal to the Fourier period ($\lambda = 1.03$ s) [30], so the scale term and the periodic term can be substituted for each other. Then the wavelet power spectrum $\left|W_n^X(s)\right|^2$ is calculated.

To eliminate edge effects (i.e., the cone of influence), we used red noise processes as the background spectrum to test the statistical significance of the wavelet power spectrum [27–29]. Values outside the cone of influence were estimated at the 95% confidence level on each scale. Correlation analyses were conducted between the time series of the primary pattern and the meteorological indices (surface air temperature, atmospheric pressure and precipitation) in January and June; $t$-tests were used to verify the statistical results.

## 3. Vertical Motion of Air over the Indian Ocean

### 3.1. Distribution of the Vertical Velocity of Air

Figure 1a shows that upward motion of atmosphere (negative) is dominant over the Indian Ocean throughout the year. The central axis of upward motion (the connection line of the upward motion center) not only moves along the meridian from north to south, but also rotates with the seasons: the central axis is at about 10° S in spring, 7.5° S in autumn and 10° S in winter. The axis rotates clockwise from winter to summer and counterclockwise from summer to winter.

There are two ascending centers in the Indian Ocean in spring and summer. In spring, the two centers are located at about (10° S, 72.5° E) and (10° S, 100° E), whereas in summer they are located at (0, 60° E) and (6° S, 90° E).

There is only one rising center in the Indian Ocean in autumn and winter. From autumn to winter, the rising center of the South Indian Ocean moves not only longitudinally, but also latitudinally. The center moves from (7.5° S, 80.5° E) in autumn to (10° S, 72° E) in winter. There is a center of subsidence in the northern Arabian Sea in spring, autumn and winter, but not in summer.

In the Bay of Bengal region of the North Indian Ocean, subsidence is dominant in winter and spring, whereas ascent is dominant in summer and autumn. This is because the Bay of Bengal is surrounded on three sides by land: the highest plateau in the world, the Tibetan Plateau, lies to the north; the Indian subcontinent lies to the west; and the Central South Peninsula lies to the east.

### 3.2. Distribution Characteristics of Atmospheric Vertical Motion over the Indian Ocean

Since an EOF analysis decomposes the original vertical velocity of the air, the spatial distribution of the principal mode represents the average distribution feature of the vertical velocity of air, and its time series represents the time-varying characteristics of the average distribution of the vertical velocity of air. Here, the explanatory variances of the principal modes for vertical motion at 850 and 500 hPa over the Indian Ocean in summer were 93% and 90%, respectively, and therefore their spatial distribution can be used to fully represent the average distribution of vertical motion at these altitudes (Figure 1b).

From Figure 1c, we know that these time coefficients are all greater than zero. This may be because we decomposed the original vertical velocity (X matrix in Equation (1)) by EOF analysis; the time series of the principal mode came to the first quadrant after EOF decomposition (coordinate rotation) [31–35]. Analysis of the principal mode showed that in summer, the vertical motion was relatively weak at 850 hPa and that there was only one center of ascending motion in the east (6° S, 90° E). The center of upward motion at 500 hPa was also located at (6° S, 90° E) (Figure 1b), which shows that the center between upper and lower layers is symmetric.

The explanatory variances in the principal modes for vertical motion at 850 and 500 hPa over the Indian Ocean in winter are 88 and 85%, respectively. There are two centers of ascending motion at 850 hPa (8° S, 61° E) and (10° S, 72° E), but only one center of ascending motion at 500 hPa (12° S, 74° E) in winter, which shows that they are asymmetric.

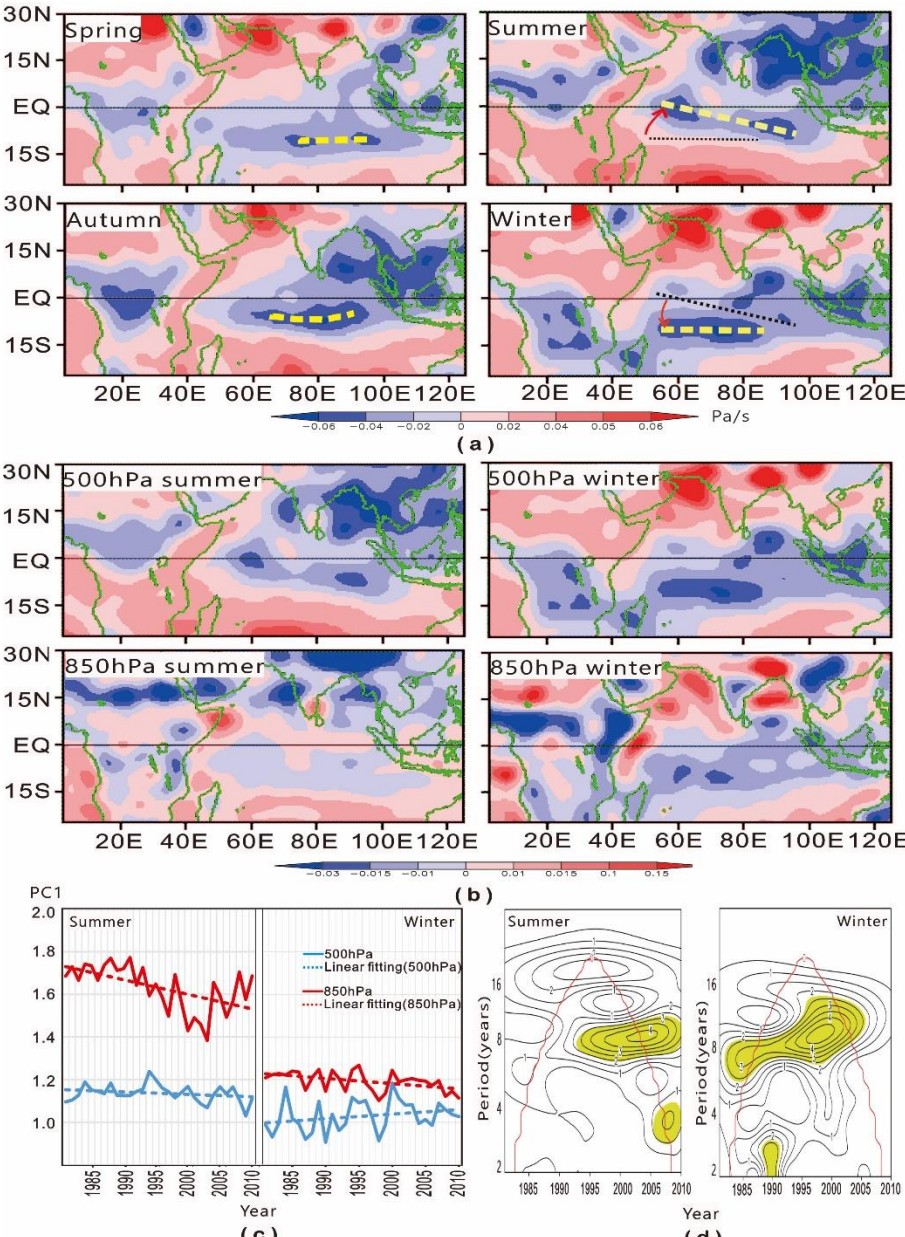

**Figure 1.** Distribution of the vertical velocity of air over the Indian Ocean. (**a**) Vertical velocity at 500 hPa in spring (March–May), summer (June–August), autumn (September–November) and winter (December–February). The yellow line delineates the central axis of upward motion. (**b**) Spatial distribution of the primary EOF-analyzed pattern for the vertical velocity at 500 and 850 hPa in summer and winter; all values passed North's significance test. (**c**) Temporal variation in the primary pattern of the vertical velocities at 500 and 850 hPa in summer and winter. (**d**) Wavelet power spectrum of the temporal coefficients of the primary pattern of the vertical velocity at 500 hPa in both summer and winter. The red line delineates the cone of influence and the yellow areas show confidence levels > 95%.

The Arabian Sea and the Bay of Bengal are dominated by the upward motion of air in summer, but by the downward motion of air in winter, as they are strongly affected by the thermal differences between the continents and the oceans.

Analysis of the time coefficients of the principal modes (Figure 1c) shows that the vertical motion at 850 and 500 hPa had a downward trend in summer from 1981 to 2010, which indicated that the vertical motion over the Indian Ocean weakened over time,

especially at 850 hPa. It has been suggested previously that global warming may weaken atmospheric motion [36]. The situation is different in winter. The vertical upward motion of air over the South Indian Ocean and the vertical subsidence of air over the North Indian Ocean both decreased at 850hPa over time. By contrast, the vertical motion at 500 hPa was enhanced—that is, the upward vertical motion of air in the South Indian Ocean and the downward vertical motion of air in the North Indian Ocean increased.

### 3.3. Period of Vertical Motion of Air

Because the explanatory variances of the principal modes at 500 hPa in summer and winter are 90 and 85%, respectively, they can be used to fully represent the distribution of the mean vertical motion of air over the Indian Ocean. We used the time coefficients of the principal mode to carry out wavelet analysis to understand the periodic variation in vertical motion over the Indian Ocean.

The distribution of the principal mode of the vertical velocity of air over the Indian Ocean in summer had a period of about 7–9 years from 1990 to 2010. The variances all passed the 95% reliability test. The periodic oscillation in winter was 9–12 years from the early 1990s to around 2003 and 2–3 years from 1987 to 1991. Both variances passed the 95% significance test (Figure 1d).

### 4. Relationship between the Vertical Motion of Air and the Climate in East Asia

The onset of the Indian monsoon in the Indian Ocean occurs in June. To understand the correlation between the climate in China and the vertical motion of air over the Indian Ocean, we analyzed the relationship between the temporal coefficients of the principal modes of the EOF analysis (explanatory variance 73%) for the vertical motion at 500 hPa over the Indian Ocean and the surface air temperature as well as the atmospheric air pressure in June. In addition, the monthly mean precipitation from the respective 839 meteorological stations in China is used to correlate with the time coefficient of the primary mode of the vertical motion of air at 500 hPa over the Indian Ocean; 839 correlation coefficients are obtained. Then, Figure 2a,b are formed by the correlation coefficients from 839 stations. For comparison, we also studied the correlation in January (explanatory variance 81%) and used the T-test to verify the correlation coefficient between them.

### 4.1. Vertical Motion of Air over the Indian Ocean and the East Asian Climate in Summer
4.1.1. Vertical Motion and Precipitation in June

The correlation between the vertical motion of air over the Indian Ocean and precipitation in China exceeded the 95% confidence level in a number of areas in June. The negative correlation in the western Tibetan Plateau, the positive correlation in the northeastern Tibetan Plateau, the negative correlation in the north of the lower reaches of Yangtze River and the positive correlation in the south of the lower reaches of Yangtze River (Figure 2a) all exceeded the 95% confidence level. These correlations indicate that with the enhancement of the vertical movement of air in the Indian Ocean in June, the precipitation decreases in the western Tibetan Plateau, increases in the northeastern Tibetan Plateau, decreases in the north and increases in the south of the lower reaches of the Yangtze River, and vice versa (Figure 2a).

4.1.2. Surface Air Temperature, Pressure and the Vertical Motion of Air in June

The vertical motion of air at 500 hPa over the Indian Ocean in June was only sporadically and positively correlated with the surface temperature and pressure over the Tibetan Plateau and northeastern China (Figure 2c,e).

### 4.2. Vertical Motion of Air over the Indian Ocean and the East Asian Climate in Winter
4.2.1. Vertical Motion of Air and Precipitation in January

Figure 2b shows that the correlation between precipitation in China and the vertical motion of air over the Indian Ocean passed the 95% significance test in five places in China

in January: there was a negative correlation with precipitation in southern Xinjiang and four positive correlations in northern Xinjiang, northeastern China, northern China and Sichuan (the upper Yangtze river), indicating that the precipitation in southern Xinjiang decreased, whereas the precipitation in the other four regions increased as the vertical motion of air over the Indian Ocean increased, and vice versa.

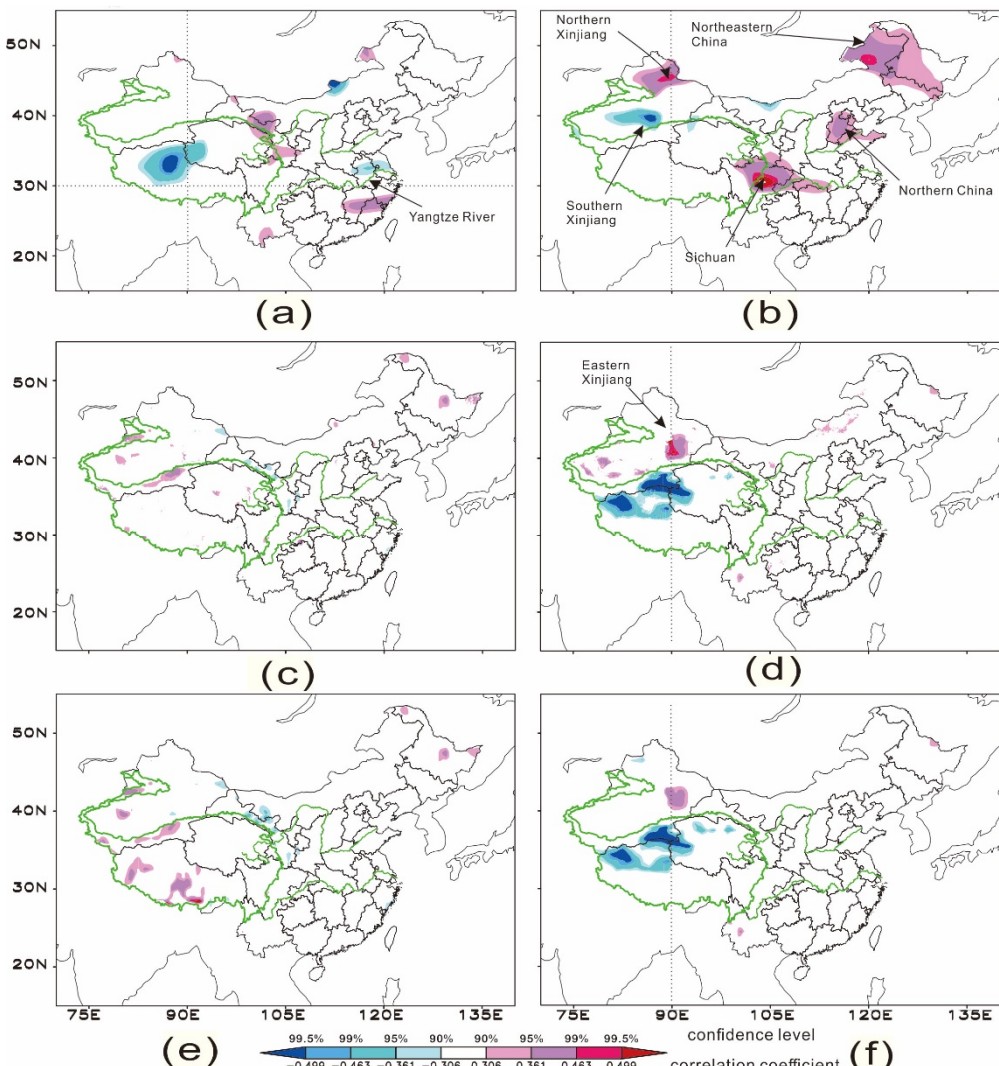

**Figure 2.** Correlation analysis between the surface meteorological variables and the vertical motion of air over the Indian Ocean. (**a**) Correlation between the precipitation and the vertical velocity in June. (**b**) Correlation between the precipitation and the vertical velocity in January. (**c**) Correlation between the surface pressure and the vertical velocity in June. (**d**) Correlation between the vertical velocity and the surface pressure in January. (**e**) Correlation between the vertical velocity and the surface temperature in June. (**f**) Correlation between the surface temperature and the vertical velocity in January.

4.2.2. Surface Air Temperature, Pressure and Vertical Motion of Air in January

The correlation between the surface air temperature in China and the vertical motion of air over the Indian Ocean in January is positive in eastern Xinjiang and negative in the northwestern Tibetan Plateau. The correlation with surface pressure is the same as that with temperature. This distribution of correlation indicates that as the vertical motion of air over the Indian Ocean increased (weakened), the surface temperature (pressure) in southern Xinjiang increased (weakened), whereas it decreased (increased) in the northwestern part Tibetan Plateau (Figure 2d,f).

## 5. Discussion

### 5.1. Relevance to Precipitation

We have shown that the precipitation in the western Tibetan Plateau is negatively correlated with the vertical motion over the Indian Ocean in June (Figure 2a). The negative correlation indicates that the precipitation in the western Tibetan Plateau decreased (enhanced) as the vertical motion over the Indian Ocean strengthened (weakened).

The average meridian circulation along 90° E in June showed that there is vertical ascending motion over both the Indian Ocean and the Tibetan Plateau (Figure 3a). These vertical motions from the Indian Ocean have sunken branches on the north of plateau, and an ascending motion below the descending branch. When the vertical motion from the Indian Ocean strengthened, the sinking motion over the north of the Tibetan Plateau is also strengthened, which suppresses the ascent of the lower layer and weakens precipitation in this area, resulting in a negative correlation.

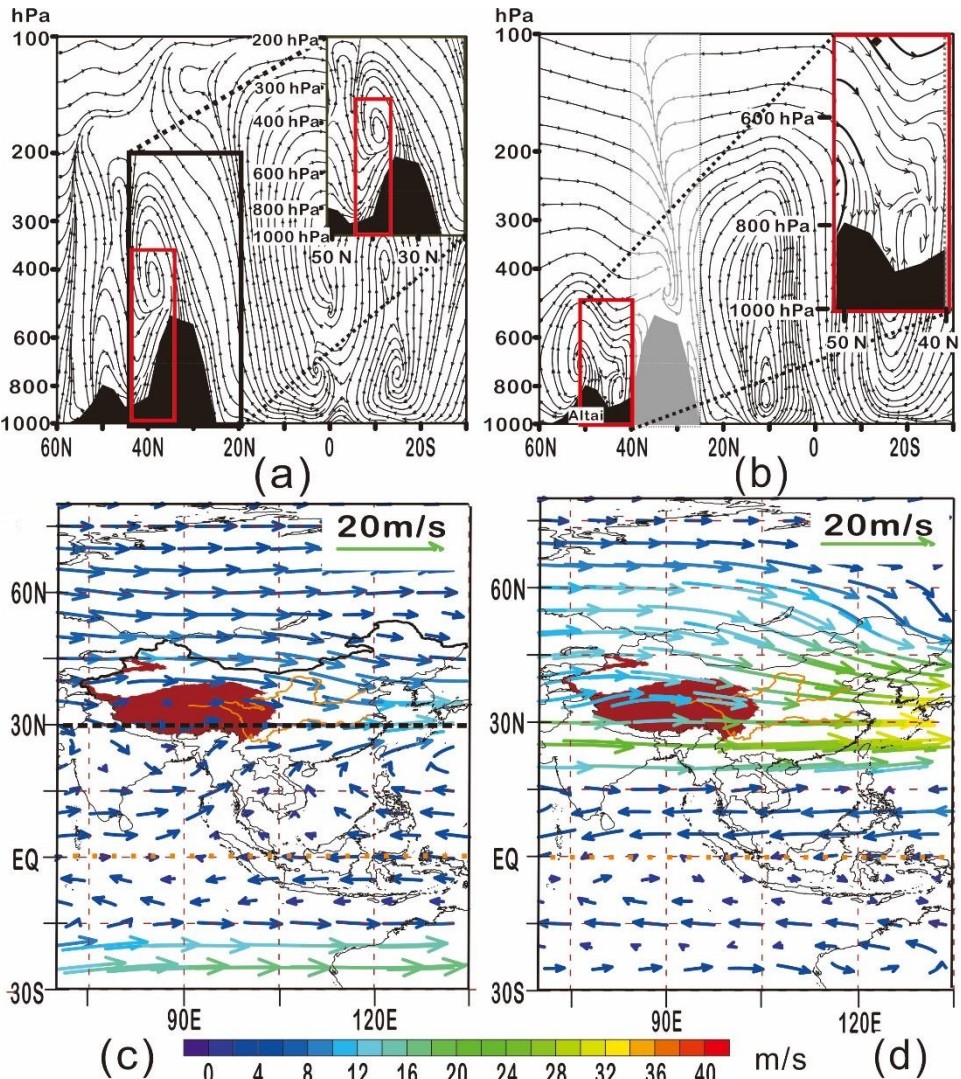

**Figure 3.** Wind fields. (**a**) Mean meridional circulation at 90° E in June. (**b**) Mean meridional circulation at 90° E in January. (**c**) Wind field at 500 hPa in June. (**d**) Wind field at 500 hPa in January. The red rectangles (**a**,**b**) represent the intersection of the updraft and the downdraft. The grey area in part (**b**) represents the Tibetan Plateau area.

Figure 2a also shows that there are two correlations between the vertical motion of air over the Indian Ocean and precipitation in the lower reaches of the Yangtze River in June. The correlation is bounded by 30° N; north of 30° N is negative correlation

and south of 30°N is a positive correlation in June. Figure 3c shows that the airflow from the Indian Ocean is also bound by 30° N, with a southwesterly airflow (summer monsoon direction) south of 30° N and a northwesterly airflow north of 30° N (in the lower reaches of the Yangtze River). The airflow is strengthened with the increase in the Indian monsoon, transporting abundant amounts of water vapor to the south of 30° N, increasing precipitation. The negative correlation to the north of 30° N (the Yangtze River) and the positive correlation between the vertical motion of air over the Indian Ocean and the precipitation in the northeastern side of the Tibetan Plateau and require further study.

In addition, we have also shown that the vertical motion over the Indian Ocean in January is negatively correlated with precipitation in southern Xinjiang and positively correlated with precipitation in northern Xinjiang, northeastern China, northern China and the Sichuan in China (the upper and middle reaches of the Yangtze River) (Figure 2b).

The negative correlation between the vertical motion of the Indian Ocean and precipitation in southern Xinjiang in January suggests that the precipitation in southern Xinjiang decreased when the vertical motion of the Indian Ocean strengthens. Figure 3b shows that cold air from the north (the high latitude) sinks over southern Xinjiang and leads to an ascending motion of the lower layer (Figure 3b). When the amount of cold air from the north (the high latitude) increases, the ascending motion in the lower layer and precipitation both increase. Cold air from the high latitude meets the air from the Indian Ocean at about 30° N, producing a strong downdraft (Figure 3b). The strengthening cold air inhibits the movement of air from the Indian Ocean, and the vertical motion over the Indian Ocean decreases—that is, precipitation in southern Xinjiang increases with the decrease in vertical motion over the Indian Ocean, leading to a negative correlation (Table 1).

**Table 1.** Correlation between climate in China and the vertical motion over the Indian Ocean in January.

| Cold Air from the High Latitude | Vertical Motion over the Indian Ocean | Precipitation in Southern/ Northern Xinjiang | Upward Motion of Air in Eastern Xinjiang | Temperature/ Pressure in the Eastern Xinjiang | Sinks in the North of the Tibetan Plateau | Temperature/ Pressure in the Tibetan Plateau |
|---|---|---|---|---|---|---|
| Strengthen (+) | Weaken (−) | Increase/decrease (+/−) | Increase (+) | Decrease/decrease (−/−) | Increase (+) | Increase/increase (+/+) |
| Weaken (−) | Strengthen (+) | Increase/decrease (−/+) | Decrease (−) | Increase/increase (+/+) | Decrease (+) | Decrease/decrease (−/−) |

The areas with a positive correlation (northern Xinjiang, northeastern China, northern China and the upper as well as the middle reaches of the Yangtze River) are all basins or plains, and they are all in the path of the cold wave that is invading China [37]. Cold air from the high latitude crosses the mountains (plateaus) before reaching the basin or plain, and then produces a downward airflow in these four areas which decreases the precipitation. The amount of precipitation decreases when the downward airflow strengthens. When cold air from the high latitude strengthens, the vertical upward motion over the Indian Ocean weakens (Table 1)—that is, when the vertical upward motion over the Indian Ocean weakens, the vertical subsidence motion over these four regions increases and the amount of precipitation decreases, leading to a positive correlation. Here, we have only given some explanation for the phenomenon we found, and there may be other reasons to be further researched. We can only show cross-sections along 90° E (Figure 3b), due to the limitation of our drawing level. The above results will be more clearly shown if a flow profile can be drawn from the origin area of high-latitude cold air to these areas with a positive correlation.

*5.2. Correlation between Vertical Motion over the Indian Ocean and Surface Temperature and Pressure in China in Winter*

Figure 2d,f has shown that there is a negative correlation between the vertical motion of air over the Indian Ocean and the surface temperature and pressure on the Tibetan Plateau in January, whereas there is a positive correlation between the vertical motion of

air over the Indian Ocean and the surface temperature and pressure in eastern Xinjiang in January. In January, deep, cold air from the high latitude moves south, and part of this package of air is overturned in the Altai Mountains and then accumulates in the northern Tibetan Plateau (eastern Xinjiang), resulting in an upward motion of air (Figure 3b). This cold air lowers the local surface temperature and pressure due to the ascent of the air. The forward movement of the cold air is then blocked by the Tibetan Plateau, and therefore sinks in the north of the plateau, increasing the surface pressure. When the sinking motion of air increases, the cloud amount decreases, the direct solar radiation increases, and the net radiation on the ground surface increases. Therefore, the radiation energy used to heat the atmosphere increases by sensible heat [38], which increases the surface temperature, and vice versa.

The cold air continues to move southward and meets the air from the Indian Ocean at about 30° N, producing a strong downdraft (Figure 3b). When the cold air from the north (high latitude) strengthens, the intersection of the two strands of air moves southward, inhibiting the vertical motion from the Indian Ocean, which then weakens. Corresponding to this weakened vertical motion over the Indian Ocean (the strengthening of cold air from the origin area of high latitude), the surface temperature and pressure decrease in eastern Xinjiang, leading to a positive correlation. As the sinking of air over northern Tibet Plateau increases, the surface pressure and temperature increase, leading to a negative correlation (Table 1).

*5.3. Turning of the Axis of Vertical Motion Center*

The axis of vertical motion in the atmosphere over the Indian Ocean turns clockwise from winter to summer, but counterclockwise back to its original position from summer to winter (Figure 1a). We conducted an EOF analysis on the anomaly in the sea–air temperature difference (sea surface temperature minus the air temperature of two meters above the sea) over the Indian Ocean from January to December. The spatial distribution from the first pattern of the anomaly rotates clockwise from January to July (the explanatory variances are 15.9% in January and 16.1% in July) and counterclockwise from summer to winter (Figure 4a,c).

We know from the wind field over the Indian Ocean at 1000 hPa that the northeasterly wind (winter monsoon) from the high latitudes of the northern hemisphere in January crosses the equator and then turns to the northwest and meets the southeasterly wind from the southern hemisphere at about 10° S (Figure 4b). In July, the southeasterly winds from the high latitudes of the southern hemisphere flow to the equator and then become southwesterly near the equator. During this process, the wind speed in the western Indian Ocean is higher than that in the eastern Indian Ocean (Figure 4d). The wind-driven currents on the surface of the Indian Ocean are affected by these changes in wind direction and wind speed. The heating field of the ocean to the atmosphere also changes with the process, and the axis of vertical motion over the Indian Ocean changes to clockwise or counterclockwise.

This study only shows the correlation between the vertical movement of air over the Indian Ocean and China's climate. The mechanism behind this phenomenon will need further study. In addition, we only calculated the monthly and seasonal mean precipitation data for 30 years from 1981 to 2010; their homogeneity was not analyzed. There are some phenomena we cannot explain yet. These deficiencies will need to be addressed in future research.

By investigating the correlation between the vertical motion of air over the Indian Ocean and surface air temperature, atmospheric pressure and precipitation in China, we found that the vertical motion of air associated with the monsoonal circulation over the Indian Ocean plays a certain role in the formation of and change in the climate in China. The results of this research have certain significance and practical value for understanding and forecasting climate in East Asia.

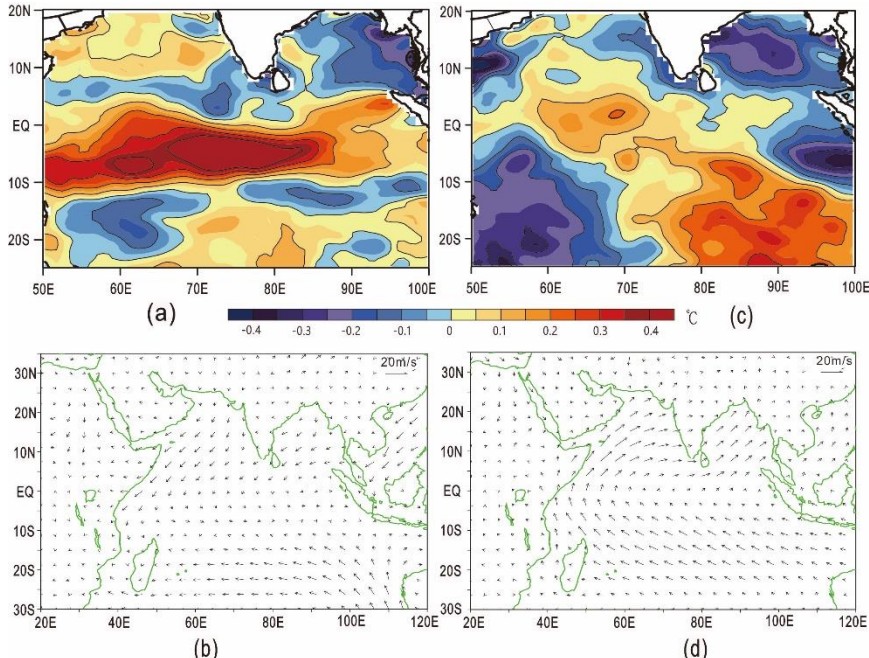

**Figure 4.** Difference in temperature between the sea and air and the wind field over the Indian Ocean. (**a**) The first pattern of the anomaly in the sea - air temperature difference (sea surface temperature minus the air temperature of two meters above the sea) in January. (**b**) Wind fields at 1000 hPa in January. (**c**) The first pattern of the anomaly in the sea-air temperature difference in July. (**d**) Wind fields at 1000 hPa in July.

## 6. Conclusions

The vertical motion of the atmosphere over the Indian Ocean is closely related to the climate in some particular regions of China. Climate diagnosis and statistical analysis were used to study the vertical motion of air over the Indian Ocean and its relationship with the climate in China. The following conclusions can be drawn from these results:

(1) The vertical motion of air is negatively correlated with precipitation in the Tibetan Plateau during summer and positively correlated with precipitation in northern Xingjiang, northeast China, northern China and the Sichuan province (the upper and middle reaches of the Yangtze River) during winter. This can be explained by the interaction between the vertical motion of air over the Indian Ocean and cold air from the high latitude of the Northern Hemisphere.

(2) The vertical motion over the Indian Ocean was weakened from 1981 to 2010, except at a height of 500 hPa in winter. The vertical motion of air over the Indian Ocean had a period of 7–9 years in summer and 2–3 in addition to 9–12 years in winter.

(3) The ascending motion of air over the Indian Ocean is dominant throughout the year. The center of ascending air moves and rotates as the seasons change, and the central axis rotates clockwise from winter to summer and counterclockwise from summer to winter. This is because the heating of the atmosphere over the Indian Ocean changes from winter to summer with the East Asian monsoon, and vice versa.

**Author Contributions:** R.T. conceived the idea and wrote the manuscript. Y.M. and W.M. revised the manuscript. All authors have read and agreed to the published version of the manuscript.

**Funding:** This research was funded by the Second Tibetan Plateau Scientific Expedition and Research Program (STEP), grant no. 2019QZKK0103, and the National Natural Science Foundation of China (grant no. 41775142).

**Institutional Review Board Statement:** Not applicable.

**Informed Consent Statement:** Not applicable.

**Data Availability Statement:** Data and methods used in the research have been presented in sufficient detail in the paper.

**Acknowledgments:** This work was supported by the Second Tibetan Plateau Scientific Expedition and Research Program (STEP), grant no. 2019QZKK0103 and the National Natural Science Foundation of China (grant no. 41775142). We acknowledge the use of meteorological data collected from the National Centers for Environmental Prediction/National Center for Atmospheric Research, the China Meteorological Administration and the Scientific Data Center for the Cold and Arid Regions of China. Duo Zha, Xinfang Zhang and Yiwei Ye completed the drawing of this paper. All data in the research can be obtained by contacting the corresponding author, Rongxiang Tian (trx@zju.edu.cn).

**Conflicts of Interest:** The authors declare no conflict of interest.

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
