# Peer review of "Vertical Motion of Air over the Indian Ocean and the Climate in East Asia"

_water, doi:10.3390/w13192641_

Round 1

Reviewer 1 Report

The paper attempts to evaluate relationships between the vertical motion of air over the Indian Ocean and the climate in East Asia, and more specifically in China. In my opinion, while the investigated problem is interesting, the submission has numerous shortcomings, which make it unsuitable for publication in the journal in present form. In general, the paper is written in a rather generalized and chaotic way, which undermines the credibility of the results obtained. More specifically, major drawbacks include:

  1. Introduction: what is the novelty and originality of the research, compared to the studies of other scientists mentioned in the literature review?
  2. P. 2, l. 91-92: “Precipitation data from 839 meteorological stations were provided by the China Meteorological Administration.” What is the time step (daily, monthly, other?) of the analyzed precipitation data? Please explain.
  3. Most of the figures are of poor quality or/and too small to read their content. For example, in Figure 1a the color scale bar is unreadable, and it also lacks the unit labeling; the same is with Figures 3 and 4.
  4. Methodology: please justify why only some regions in China were selected for the analysis, instead of the entire monsoon China.
  5. Methodology: I don’t understand what specifically precipitation data were correlated with the vertical motion of air. Did you correlate precipitation totals form each of the respective 839 meteorological stations or maybe you averaged some datasets collected in regions separated in you study? The analyzed regions (as Xinjiang, North-Eastern China, Northern China, Sichuan and other) cover hundreds of thousands square kilometres, and I doubt if in each of these regions the correlations were purely positive or only negative.
  6. P. 11, l. 403-404: “In addition, we only calculated the monthly and seasonal mean precipitation data for 30 years from 1981 to 2010, their homogenity was not analyzed.” Information on the time step should be put in the description of the “2.1. Data” sub-section (please refer to the aforementioned remark No. 2). As for checking the data homogeneity: this is one of the most basic and initial requirement of any research, and without it the credibility of any study is very doubtful. So, in fact your study should begin with the homogeneity analysis.
  7. There is lack of the “Discussion” chapter, in which you should - among others - point on the practical value of the findings of your study.
  8. The list of references needs to be reformatted, in accordance with the “Information for Authors”.
  9. The paper requires numerous linguistic improvements.

Generally, it is recommended to accept the paper for publication after substantial improvements.

Reviewer 2 Report

The manuscript analyses the vertical motion of air over the Indian Ocean and its possible relation with the climate in East Asia. Overall, the paper is interesting and I think that it has the potential for publication in Water after minor revisions, considering the list of comments below:

  • The English should be improved and I recommend a professional proofreading by a native speaker.
  • Please clearly specify the aims of the paper (lines 65-80).
  • Figures 1a, 1b and 2: a bigger legend is needed.
  • Figure 1c: Use the same division for the Y-axis
  • Figures 3a and b: put the latitude on the Y-axis.

Round 2

Reviewer 1 Report

Generally, the Authors have made satisfactory improvements in the paper, according to the reviewer's remarks. However, linguistic corrections are still necessary.